# Anti-Müllerian hormone and fertility in women after childhood cancer treatment: Association with current infertility risk classifications

**Anna Nyström**[1,2¤a]*, **Helena Mörse**[1,3¤a], **Ingrid Øra**[1¤a], **Emir Henic**[4,5¤b], **Jacob Engellau**[6,7¤a], **Elinore Wieslander**[7], **Andrzej Tomaszewicz**[7], **Maria Elfving**[1,8¤a]

1 Department of Clinical Sciences Lund, Paediatrics, Lund University, Lund, Sweden, 2 Paediatric Cardiology, Skåne University Hospital, Lund, Sweden, 3 Paediatric Oncology and Haematology, Skåne University Hospital, Lund, Sweden, 4 Department of Translational Medicine, Reproductive Medicine, Lund University, Malmö, Sweden, 5 Reproductive Medicine, Skåne University Hospital, Malmö, Sweden, 6 Department of Clinical Sciences Lund, Systemic Radiation Therapy, Lund University, Lund, Sweden, 7 Haematology, Oncology and Radiation Physics, Skåne University Hospital, Lund, Sweden, 8 Paediatric Endocrinology, Skåne University Hospital, Lund, Sweden

¤a Current address: Department of Clinical Sciences Lund, Lund University, Lund, Sweden
* anna.nystrom@med.lu.se

**Data Availability Statement:** The data underlying this article is not publicly available due to the General Data Protection Regulation (GDPR, a

## Abstract

### Background

To identify childhood cancer survivors (CCSs) at risk of premature ovarian insufficiency (POI) and impaired fertility is important given its impact on quality of life. The aim of this study was to assess ovarian markers and fertility outcomes in adult female CCSs. We used the Swedish and the PanCareLIFE classifications for infertility risk grouping.

### Methods

167 CCSs, at median age 34.6 years (19.3–57.8) with a median follow-up time of 25.4 years (11.6–41.3), and 164 healthy matched controls were included in this cross-sectional study. We assessed anti-Müllerian hormone (AMH) levels, antral follicle count (AFC), ovarian volume (OV), and fertility outcomes. Based on gonadotoxic treatments given, CCSs were categorized into infertility risk groups.

### Results

The median levels of AMH, AFC and OV were lower in CCSs (1.9 vs. 2.1 ng/ml, 12.0 vs. 13.0, 6.8 vs. 8.0 cm$^3$) compared with controls, although statistically significant only for OV ($p = 0.021$). AMH levels in CCSs <40 years were lower for those classified as high-risk ($p = 0.034$) and very high-risk ($p<0.001$) for infertility, based on the Swedish risk classification. Similarly, AFC was reduced in the high-risk ($p<0.001$) and the very high-risk groups ($p = 0.003$). CCSs of all ages showed a trend towards impaired fertility, especially in the very high-risk group. POI was diagnosed in 22/167 CCSs, of whom 14 were in the high- and very high-risk groups. The results according to the PanCareLIFE classification were similar.

common European Union regulation), which regulates the handling of personal data and privacy protection in Sweden. Even though the data is pseudonymised, it can potentially be identifying with additional sensitive information (e.g., specific dates, cancer diagnoses and treatments, etc.), and therefore considered personal data under the GDPR. However, some data might be shared on reasonable request to Lund University (contact via ikvl-prefektadmin@med.lu.se) if the receiver can guarantee a high level of data protection.

**Funding:** This study was financed by the Swedish Childhood Cancer Foundation (https://www. barncancerfonden.se/), the Regional Funding of Skåne (https://sodrasjukvardsregionen.se/), and Skåne University Hospital Donation Fund (https:// vard.skane.se/en/skane-university-hospital/), with grant numbers DK2023-0004, 2022-1342, and 2022-926, respectively. All the awards were received by ME. The funders had no role in study design, data collection and analysis, decision to publish, or preparation of the manuscript.

**Competing interests:** The authors have declared that no competing interests exist.

## Conclusion

Both the Swedish and the PanCareLIFE infertility risk classifications are reliable tools for identifying those at risk of reduced ovarian markers and fertility, as well as POI. We recommend fertility preservation counselling for patients receiving highly gonadotoxic treatments (i.e., Cyclophosphamide Equivalent Dose $\geq 6$ g/m$^2$, radiotherapy exposure to ovaries or stem cell transplantation) with follow-up at a young reproductive age due to the risk of a shortened reproductive window.

## Introduction

As the current 5-year survival for childhood cancer has reached 80% in Europe [1], health care providers need to manage late side effects such as ovarian dysfunction and impaired fertility. Endocrine complications are one of the most common side effects, affecting nearly 50% of survivors, occurring shortly after treatment or over time [2, 3]. It is known that alkylating agents, radiotherapy, and conditioning regimen before stem cell transplantation are all highly gonadotoxic and may induce follicular loss leading to premature ovarian insufficiency (POI) [4–6]. The definition of POI is amenorrhea below the age of 40 years with low levels of oestrogen and elevated levels of follicle-stimulating hormone (FSH) [7]. Furthermore, POI is associated with infertility, osteoporosis, cardiovascular disease, metabolic syndrome, and impaired quality of life [7–9].

Anti-Müllerian hormone (AMH) is a reliable marker for assessment of ovarian reserve in adult females. Previously, we reported that AMH correlates well with the antral follicle count (AFC) both in female childhood cancer survivors (CCSs) and in controls [10]. It is produced by the granulosa cells of immature follicles to the antral stage [11]. AMH is measurable at birth, whereafter it increases and peaks at age 24.5 years. Following this peak, AMH declines and is no longer detectable at menopause [12]. The levels of AMH throughout the menstrual cycle are believed to be fairly stable, in contrast to other serum markers of ovarian function, thus are assessable regardless of the cycle phase [13, 14]. AMH also has the advantage of being a more sensitive marker to detect impaired ovarian function as it declines before FSH levels rise [15, 16].

Patients requiring highly gonadotoxic treatment can be offered fertility preservation. Oocyte cryopreservation is an established method which should be offered to post-pubescent girls at high infertility risk if not delaying cancer treatment, as stated both in the Swedish and the PanCareLIFE guidelines for infertility risk assessment [17, 18]. For pre-pubescent girls, however, this method is not possible due to immature oocytes. The only method available for these girls is ovarian tissue cryopreservation for subsequent reimplantation, which until recently was considered experimental [17]. The above-mentioned guidelines differ somewhat for pre-pubescent girls receiving highly gonadotoxic treatment as the Swedish guidelines propose ovarian tissue cryopreservation, while the PanCareLIFE guidelines provide only moderate support for this intervention [17, 18]. Moreover, cancer survivors report psychological distress due to impaired reproductive health after cancer treatment [19]. On the other hand, women who had infertility counselling and fertility preservation before treatment report an improved quality of life [20]. Therefore, identification of treatment risk factors for potential infertility is essential for appropriate counselling and fertility preservation to secure future reproduction in these patients.

The aim of this cross-sectional study was to investigate the late side effects of childhood cancer treatment on ovarian function and fertility in adult CCSs. We evaluated the current

infertility risk assessments as stated in the Swedish and the PanCareLIFE recommendations. These aims were achieved.

## Materials and methods

### Subjects

The study included female CCSs treated for childhood cancer in southern Sweden. CCSs had been diagnosed with cancer under 18 years of age, received treatment between 1964 and 2008, and had completed treatment more than two years prior. Those with rare cancer diagnoses (i.e., skin and thyroid cancer) or solid tumours outside the central nervous system treated surgically only were excluded. CCSs and controls eligible for the study were identified through the Swedish Cancer Registry and Swedish Population Registry, respectively. Controls were matched regarding sex, birth date, ethnicity, and smoking habits.

The included participants completed questionnaires concerning ongoing and earlier hormonal treatment, number of pregnancies, children born, adopted children, as well as desire for future children. Information about primary or secondary amenorrhea, and fertility investigation and treatment were retrieved. Detailed information was collected, regarding cancer diagnoses and treatment, from the Childhood Cancer Registry, BORISS (Paediatric Oncology Registry in South Sweden) [21], and medical records. The cumulative doses of chemotherapeutic agents and radiotherapy given were retrieved for each patient. For this study, the recruitment period was between 15 October 2010 and 30 November 2015.

CCSs were categorized to four risk groups according to the current Swedish strategy of treatment risk grouping for infertility (Table 1) [18]. The recommendations for infertility risk grouping were recently updated with a lower cumulative cyclophosphamide dose (i.e., >6 g/m$^2$) for the high-risk group. Those not matching the four risk groups were allocated to the following groups: no risk, surgery only, or unilateral oophorectomy. In addition, CCSs were divided into four infertility risk groups according to the PanCareLIFE recommendations (Table 2) [17]. As the PanCareLIFE recommendations are based on Cyclophosphamide Equivalent Dose (CED), a quantitative measure of cumulative alkylating agent exposure, we calculated CED according to the following equation: CED (mg/m$^2$) = 1.0 (cumulative cyclophosphamide dose [mg/m$^2$]) + 0.244 (cumulative ifosfamide dose [mg/m$^2$]) + 0.857 (cumulative procarbazine dose [mg/m$^2$]) + 14.286 (cumulative chlorambucil dose [mg/m$^2$]) + 15.0 (cumulative BCNU dose [mg/m$^2$]) + 16.0 (cumulative CCNU dose [mg/m$^2$]) + 40 (cumulative melphalan dose [mg/m$^2$]) + 50 (cumulative Thio-TEPA dose [mg/m$^2$]) + 100 (cumulative nitrogen mustard dose [mg/m$^2$]) + 8.823 (cumulative busulfan dose [mg/m$^2$]) [22]. To our knowledge, there are no studies

**Table 1. Swedish treatment risk classification of infertility in girls.**

| Low-risk | Moderate-risk | High-risk | Very high-risk |
|---|---|---|---|
| • Vincristine | • Cisplatin | • Cyclophosphamide >6 g/m$^2$ | • >10 Gy to the ovaries |
| • Methotrexate | • Carboplatin | • Ifosfamide >60 g/m$^2$ | • Allogenic HSCT |
| • Actinomycin D | • Cyclophosphamide <6 g/m$^2$ | • Procarbazine | • Autologous HSCT |
| • Bleomycin | • Ifosfamide <60 g/m$^2$ | • BCNU | |
| • Mercaptopurine | • CCNU <360 mg/m$^2$ | • CCNU >360 mg/m$^2$ | |
| • Vinblastine | | • <10 Gy to the ovaries | |
| • 5-fluorouracil (5-FU) | | | |

Gy: gray; HSCT: hematopoietic stem cell transplantation. This is the forthcoming version with updated cumulative doses for cyclophosphamide (previously, <9 g/m$^2$ and >9 g/m$^2$ for the moderate- and high-risk groups, respectively) (Vävnadsrådet [Unpublished]).

**Table 2. Treatment risk classification of infertility in girls based on the PanCareLIFE recommendations.**

| Group 1 | Group 2 | Group 3 | Group 4 |
|---|---|---|---|
| Other treatments | Unilateral oophorectomy | • CED <6 g/m$^2$ | • CED $\geq$6 g/m$^2$ |
| | | • Cranial radiotherapy | • Ovarian radiotherapy* |
| | | | • HSCT |

Group 1 is not considered at risk for infertility, while groups 2–4 have a potential risk for infertility.

CED: cyclophosphamide-equivalent dose; HSCT: hematopoietic stem cell transplantation.

*irrespective of ovarian radiotherapy dose.

comparing similar intravenous and oral doses of cyclophosphamide, whereas the latter was neither included in CED nor the Swedish risk assessment. In addition, we defined alkylating agents as described by van Dorp et al. [23], although several of these agents are not included in CED (i.e., carboplatin, cisplatin, mustine, and dacarbazine).

## Ethical approval

The study was approved by the Regional Ethics Committee, Medical Faculty, Lund University, Sweden (approval no. 523/2009). A written informed consent was obtained from all participants prior to inclusion.

## Estimation of radiotherapy dose to the ovaries

Radiotherapy doses to the right and left ovary were estimated for those treated with abdominal radiotherapy. For 21 CCSs who received craniospinal or flank radiotherapy, the dose to each ovary was extracted by a radiation oncologist and two radiation physicists from individual radiotherapy charts including x-ray images and/or photographs/schematical anatomical drawings of the treatment fields. The lateral and superior/inferior location of the ovary was based on Bardo et al. and the ventral/dorsal location was approximated as the midline [24]. Due to the uncertainty of the ovaries' position in relation to the radiation fields, the dose was in most cases reported as an interval. If there was a difference in radiotherapy doses between ovaries, the patient was categorized according to the lowest dose to one ovary.

## Ultrasound

The clinical examination was performed between October 2010 and January 2015 at the Reproductive Medicine Centre (RMC) at Skåne University Hospital, Malmö, Sweden. Transvaginal ultrasound was used for assessment of AFC and ovarian volume (OV). AFC is presented as the sum of follicles with diameter 2–10 mm on both sides and OV as the total volume of the ovaries. Six different doctors conducted the examination with the BK Medical Flex Focus 500 and BK Medical Pro Focus Scanner.

## Hormonal assay

Fasting blood samples for AMH measurement were obtained during the early follicular phase (i.e., day 2–5 of the menstrual cycle) for females with regular cycles, and at a random day for those with irregular cycles, amenorrhea, or ongoing treatment with hormone replacement therapy (HRT) or oral contraceptives (OC). Before analysis, blood samples were centrifuged and frozen to -20˚C. Serum AMH levels were analysed using ultrasensitive enzyme-linked immunosorbent assay (ELISA) at the Laboratory of Reproductive Biology, Copenhagen,

Denmark. The detection limit was 0.023 ng/ml and values <0.023 ng/ml were specified as 0. The intra- and inter-assay variation coefficients were <10%.

Serum levels of inhibin B, FSH and 17β-oestradiol were analysed but not presented in this study as we previously reported that AMH is the most accurate serum marker for ovarian reserve post childhood cancer treatment [10].

## Diagnosis of POI

The former POI definition was not used as almost all females with ovarian insufficiency were on HRT, thus affecting levels of FSH and 17β-oestradiol. Instead, POI was defined as very low (<0.1 ng/ml) or undetectable AMH with a history of amenorrhea and treatment with HRT below the age of 40 years [10].

## Statistical analysis

Descriptive data are presented as count, median, range and percentage. We used histograms to assess the distribution of the data. Mann-Whitney U test, Pearson correlation, partial correlation and linear regression were performed for continuous variable analyses. Scatter plots illustrate correlations of various variables. The Kruskal-Wallis test and the Bonferroni correction were applied when comparing more than two groups. For analysis of categorical variables, the Fisher's exact test and chi-squared test were used. Statistical tests were two-sided. The level of significance was set at $p<0.05$ and the confidence interval (CI) at 95%. All the analyses were performed with the IBM Statistical Package for the Social Science (SPSS) version 26.

## Results

### Participants—Diagnoses and treatments

From the Swedish Cancer Registry 575 females met the inclusion criteria, of whom 244 were excluded due to cancer in situ, cervical cell atypia, or no mature cancer (i.e., diagnoses generally not represented in the childhood cancer population). 202 females accepted participation, of whom 28 dropped out due to lack of time, three with ongoing pregnancy and four with pronounced disabilities were excluded, resulting in a total of 167 CCSs in this study [8]. We recruited initially 167 matched controls, and three dropped out, leaving 164 controls included in the study.

Offspring data of both participants and nonparticipants were collected from the Swedish Multi-Generation Registry at the National Board of Social Affairs and Health for drop-out analysis. The participants were representative of the total CCSs group, as no great differences regarding number of offspring were noted for those included and among those who declined or did not respond [10].

The distribution of childhood cancer diagnoses for CCSs was representative of children with cancer between 1984 and 2010 in Sweden [25]. Cancer diagnoses and treatments are demonstrated in S1 Table. In total, 127 (76%), 81 (49%) and 11 (7%) were treated with chemotherapy, alkylating agents, and hematopoietic stem cell transplantation (HSCT), respectively. 87 (52%) underwent radiotherapy, of whom 53 (32%) had cranial, 34 (20%) abdominal, 16 (10%) cranial and abdominal, and 7 (4%) had total body irradiation. 19 (11%) of CCSs had surgery as the only treatment. The main characteristics for CCSs and controls are presented in Table 3.

**Table 3. Background data for childhood cancer survivors (CCSs) and controls.**

|  | CCSs n = 167 | Controls n = 164 | *p*-value |
|---|---|---|---|
| Age at examination (yr) | 34.6 (19.3–57.8) | 35.8 (19.3–58.0) | 0.385 |
| Age at diagnosis (yr) | 8.4 (0.1–17.9) | n.a. | n.a. |
| Time since diagnosis (yr) | 25.4 (11.6–41.3) | n.a. | n.a. |
| Height (cm) | 164.5 (143.0–181.5) | 168.5 (150.0–186.4) | <**0.001** |
| Weight (kg) | 65.4 (41.0–125.0) | 66.2 (46.6–107.2) | 0.827 |
| Body mass index (kg/m$^2$) | 24.1 (16–44) | 22.9 (18–35) | **0.027** |
| HRT | n = 20 (12%) | n = 0 | <**0.001** |
| OC (including p-ring and systemic gestagens/progestins) | n = 48 (29%) | n = 58 (35%) | 0.239 |
| POI (primary) | n = 22 (13%) | n = 0 | <**0.001** |
| Hypothalamic-pituitary-ovarian insufficiency | n = 5 (3%) | n = 0 | n.a. |

Characteristics shown as median, range, count and percentage. Three CCSs were on OC as HRT and are counted in for both groups.

yr: years; HRT: hormone replacement therapy; OC: oral contraceptives; POI: premature ovarian insufficiency; n.a.: not applicable.

*P*-value calculated with Mann-Whitney U test and Fisher's exact test.

## Ovarian markers—AMH, AFC and OV

For the analysis of AMH, 166 CCSs and 163 controls were included. AFC could be assessed for 135 CCSs and 157 controls, and OV for 129 CCSs and 155 controls. If the total AFC and OV were non-measurable (i.e., due to oophorectomy [unilateral or bilateral], ovarian cysts, non-imaging, or virginity), the participants were excluded from the analysis. AFC was measurable in some participants despite non-assessable OV due to follicular cysts.

Median levels of AMH, AFC and OV were lower in CCSs (1.9 vs. 2.1 ng/ml, 12.0 vs. 13.0, 6.8 vs. 8.0 cm$^3$) compared with controls, but statistically significant for OV only ($p$ = 0.021). There was no significant difference in median AMH levels between CCSs (n = 65) and controls (n = 57) on OC or HRT: 1.8 (0.0–23.5) and 2.4 (0.0–14.3) ng/ml, respectively ($p$ = 0.077). In females below 40 years of age, there was no difference in median AMH levels for CCSs 3.3 ng/ml (n = 120) and controls 3.2 ng/ml (n = 112) ($p$ = 0.108). Median AFC was 13.0 among CCSs (n = 102) and 15.0 among controls (n = 109) ($p$ = 0.026). Furthermore, median OV was lower in CCSs (n = 98) compared with controls (n = 108), 7.6 and 9.1 cm$^3$, respectively ($p$ = 0.009).

AMH levels were negatively correlated with age at examination among both CCSs (r = -0.435, $p$<0.001, n = 166) and controls (r = -0.497, $p$<0.001, n = 163). When comparing CCSs and controls with linear regression analysis, CCSs had slightly lower AMH levels at all ages, but the difference was not significant ($p$ = 0.224) and was not more pronounced with increasing age (Fig 1).

## Swedish infertility risk groups

CCSs were divided into four infertility risk groups based on cancer treatment given, as shown in Table 1. Furthermore, we included the remaining CCSs not belonging to the four risk groups in three groups: no risk (i.e., no treatment according to Table 1), surgery only, or unilateral oophorectomy. For the unilateral oophorectomy group, three, three, and five CCSs had undergone treatment corresponding to the very high-risk, high-risk, and moderate-risk groups, respectively. The remaining two were treated with the alkylating agent Melphalan as the only treatment except for surgery. Three CCSs in the unilateral oophorectomy group

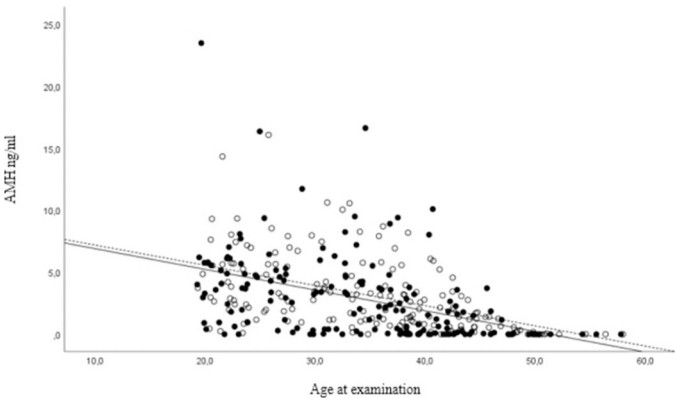

**Fig 1. Correlation between AMH levels and age at examination.** Childhood cancer survivors (CCSs) = ●+filled line (n = 166), controls = ○+dashed line (n = 163). AMH levels showed a negative correlation with age at examination among both CCSs (r = -0.435, *p*<0.001) and controls (r = -0.497, *p*<0.001). AMH: anti-Müllerian hormone.

underwent oophorectomy which was not related to their cancer diagnosis (i.e., endometriosis, myoma and unknown reason). The main results for the infertility risk groups for all CCSs and controls are presented in Table 4.

Altogether, 31 CCSs were allocated to the low-risk group, 29 to the moderate-risk group, 37 to the high-risk group, 13 to the very high-risk group, 17 to the no risk group, 19 to the surgery only group, and 13 to the unilateral oophorectomy group. For six CCSs the risk assessment was not available due to inconclusive data, and another two females with bilateral oophorectomy were not included in any subgroup. The reference group consisted of a total of 164 healthy controls. There was a difference in age at examination across the groups analysed with the Kruskal-Wallis test (*p* = 0.015), but pairwise comparisons showed no difference (Table 4). The low- and moderate-risk groups were younger at diagnosis with a median of 5.0 and 6.1 years, respectively. For the remaining groups, median age at diagnosis was higher, especially for the very high-risk, unilateral oophorectomy, and no risk groups. The Kruskal-Wallis test

**Table 4. Ovarian markers in childhood cancer survivors (CCSs) (n = 167) and controls (n = 164).**

| | No risk | Low-risk | Moderate-risk | High-risk | Very high-risk | Only surgery | Unilateral oophorectomy | All CCSs | Controls | *p*-value All CCSs vs controls | Overall *p*-value |
|---|---|---|---|---|---|---|---|---|---|---|---|
| N | 17 | 31 | 29 | 37 | 13 | 19 | 13 | 167 | 164 | n.a. | n.a. |
| Age at examination (years) | 32.9 (19.6–49.8) | 31.8 (19.8–48.0) | 28.8 (19.5–47.9) | 36.6 (19.3–57.8) | 30.8 (21.8–51.4) | 38.1 (22.2–49.9) | 39.4 (25.0–54.3) | 34.6 (19.3–57.8) | 35.8 (19.3–58.0) | 0.385 | **0.015** |
| Age at diagnosis (years) | 11.1 (1.3–17.9) | 5.0 (0.6–16.0) | 6.1 (0.1–17.2) | 7.9 (0.4–17.6) | 12.3 (1.8–17.9) | 8.8 (0.1–17.2) | 12.6 (4.6–17.0) | 8.4 (0.1–17.9) | n.a. | n.a. | **0.003** |
| AMH (ng/ml) | 2.2 (0.0–23.5) | 4.0 (0.0–8.9) | 3.7 (0.0–16.6) | 0.8 (0.0–10.1) | 0.0 (0.0–4.9) | 3.1 (0.0–9.4) | 0.0 (0.0–16.4) | 1.9 (0.0–23.5) | 2.1 (0.0–16.1) | 0.065 | **<0.001** |
| AFC | 8.0 (0–47) | 15.0 (4–28) | 18.0 (3–38) | 7.0 (0–23) | 1.5 (0–16) | 14.0 (0–26) | n.a. | 12.0 (0–47) | 13.0 (0–40) | 0.096 | **<0.001** |
| OV (cm³) | 6.0 (0.9–15.4) | 6.6 (1.7–21.4) | 8.1 (2.2–20.2) | 5.5 (0.7–20.5) | 1.9 (0.3–21.0) | 7.0 (1.8–15.5) | n.a. | 6.8 (0.3–21.4) | 8.0 (0.9–36.7) | **0.021** | **0.028** |

CCSs were divided into Swedish infertility risk groups according to cancer treatment given (Table 1).

For six CCSs risk assessment was not available; two with bilateral oophorectomy were not included.

showed a significant difference in age at diagnosis ($p$ = 0.003). Furthermore, for pairwise comparisons there was a significant difference in age at diagnosis between the low-risk and unilateral oophorectomy groups ($p$ = 0.006).

For AMH, AFC, and OV there were differences between groups: $p<0.001$, $p<0.001$, and $p$ = 0.028, respectively (Table 4). Moreover, AMH levels were significantly lower in the very high-risk group compared with controls ($p<0.001$). AFC was reduced for both the very high-risk and high-risk groups compared with controls ($p$ = 0.007 and $p$ = 0.012, respectively). There was a tendency for decreased OV in the very high-risk group compared with controls.

## Swedish infertility risk groups <40 years

As AMH decreases with increased age, we analysed 118 CCSs and 113 controls below 40 years of age separately in the different infertility risk groups. Bilateral oophorectomy was an exclusion criterion, although no females <40 years of age underwent such treatment. Furthermore, two CCSs were excluded as radiotherapy doses to the ovaries could not be estimated. The results for the infertility risk groups and controls <40 years are presented in Table 5.

The discrepancy for age at examination remained ($p$ = 0.049), yet again no difference was noted with pairwise comparisons. Those with low and moderate risk still had lower median age at diagnosis: 3.5 and 3.4 years, respectively. The Kruskal-Wallis test showed a significant difference for age at diagnosis ($p$ = 0.007), but no difference was observed when pairwise comparisons were performed.

AMH and AFC differed across the groups, $p<0.001$ for both comparisons (Table 5). When pairwise comparisons were used, AMH levels were significantly lower in the very high-risk group compared with controls ($p<0.001$). Moreover, the high-risk group had decreased AMH levels compared with controls ($p$ = 0.034) (Fig 2a). AFC was significantly reduced when

**Table 5. Ovarian markers in childhood cancer survivors (CCSs) (n = 120) and controls (n = 113) <40 years.**

| | No risk | Low-risk | Moderate-risk | High-risk | Very high-risk | Only surgery | Unilateral oophorectomy | All CCSs <40 years | Controls <40 years | p-value All CCSs vs controls <40 years | Overall p-value |
|---|---|---|---|---|---|---|---|---|---|---|---|
| N | 12 | 26 | 24 | 28 | 10 | 11 | 7 | 120 | 113 | n.a. | n.a. |
| Age at examination (years) | 30.3 (19.6–37.4) | 28.5 (19.8–37.3) | 25.3 (19.5–38.6) | 33.7 (19.3–39.5) | 30.3 (21.8–39.8) | 26.4 (22.2–38.3) | 38.3 (25.0–39.4) | 30.6 (19.3–39.8) | 30.8 (19.3–39.9) | 0.599 | **0.049** |
| Age at diagnosis (years) | 10.2 (1.3–15.6) | 3.5 (0.6–12.0) | 3.4 (0.1–17.0) | 8.1 (0.4–17.2) | 9.7 (1.8–17.2) | 6.7 (0.1–15.1) | 12.5 (4.6–15.0) | 6.4 (0.1–17.2) | n.a. | n.a. | **0.007** |
| AMH (ng/ml) | 3.3 (0.5–23.5) | 4.3 (0.4–8.9) | 4.0 (0.2–16.6) | 1.1 (0.0–7.2) | 0.0 (0.0–4.9) | 3.7 (0.1–9.4) | 0.1 (0.0–16.4) | 3.3 (0.0–23.5) | 3.2 (0.0–16.1) | 0.108 | **<0.001** |
| AFC | 13.5 (3–47) | 14.0 (4–28) | 19.0 (3–38) | 9.0 (0–23) | 3.0 (0–16) | 21.0 (10–26) | n.a. | 13.0 (0–47) | 15.0 (5–40) | **0.026** | **<0.001** |
| OV (cm³) | 6.8 (1.5–15.4) | 6.2 (1.7–21.4) | 8.1 (2.2–20.2) | 6.0 (0.7–20.5) | 5.5 (0.5–21.0) | 10.6 (3.2–15.5) | n.a. | 7.6 (0.25–21.4) | 9.1 (1.6–36.7) | **0.009** | 0.054 |

CCSs were divided into Swedish infertility risk groups according to cancer treatment given (Table 1).

For two CCSs risk assessment was not available.

Data presented as count, median, and range.

AMH: anti-Müllerian hormone; AFC: antral follicle count; OV: ovarian volume; n.a.: not applicable.

$P$-value for all CCSs vs controls calculated with Mann-Whitney U test.

Overall $p$-value calculated with Kruskal-Wallis test.

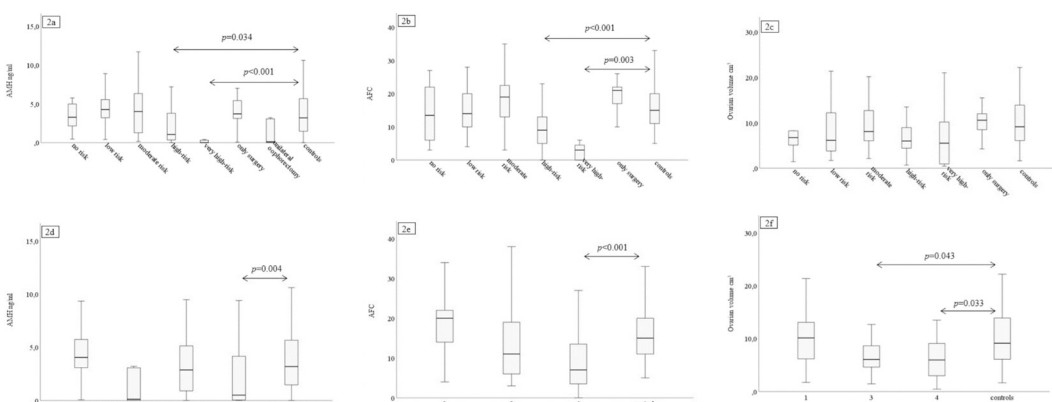

**Fig 2.** a-f. Pairwise comparisons between Swedish infertility risk groups (2a–2c), PanCareLIFE infertility risk groups (2d–2f), and controls for AMH, AFC, and OV, <40 years. 120 CCSs and 113 controls were included. For the Swedish infertility risk groups, two CCSs were excluded due to inconclusive radiotherapy dose to the ovaries. Outliers are not presented. AMH: anti-Müllerian hormone; AFC: antral follicle count; OV: ovarian volume. The Kruskal-Wallis test and the Bonferroni correction were used.

comparing the very high-risk group, as well as the high-risk group, with controls ($p = 0.003$ and $p<0.001$, respectively) (Fig 2b). There was no difference across the groups regarding OV ($p = 0.054$) (Table 5 and Fig 2c).

The high-risk group had 1.7 ng/ml lower AMH levels compared with controls when analysed with linear regression ($p = 0.012$, 95% CI -3.0 –-0.4, adjusted for age at examination). Levels of AMH were reduced 3.5 ng/ml in the very high-risk group compared with controls ($p = 0.001$, 95% CI -5.6 –-1.4, adjusted for age at examination). The unadjusted results were similar.

## PanCareLIFE infertility risk groups

CCSs were also divided into four infertility risk groups according to the PanCareLIFE recommendations, with group 4 receiving the most gonadotoxic treatment (i.e., CED $\geq$6 g/m$^2$, ovarian radiotherapy, or HSCT) (Table 2). Group 3 received lower CED (<6 g/m$^2$) or cranial radiotherapy. CCSs who underwent unilateral oophorectomy were assigned to group 2 despite other treatments given. Five, four and four females in group 2 were given treatment that matched that of groups 4, 3 and 1, respectively. Group 1 consisted of CCSs with other treatments than the above mentioned. It should be noted that one CCS in group 3 had upper abdominal radiotherapy and was therefore not assigned to group 4 (S1 Table). Altogether, 165 CCSs were included as two CCSs with bilateral oophorectomy were excluded. The main results for the groups at all ages are shown in Table 6.

There was no significant difference across the groups regarding age at examination ($p = 0.053$). For age at diagnosis there was a significant difference between groups 1 and 2, 6.2 and 12.6 years, respectively ($p = 0.015$). Median AMH levels were significantly lower in groups 2 and 4 compared with controls: 0.0 and 0.4 vs. 2.1 ng/ml, respectively ($p = 0.018$ and $p = 0.004$, respectively). Both AFC and OV were significantly reduced when comparing group 4 with controls ($p = 0.001$ and $p = 0.040$, respectively).

## PanCareLIFE infertility risk groups <40 years

For the analysis of women <40 years, we included 120 CCSs and 113 controls (Table 7). CCSs in group 1 were significantly younger at examination compared with controls: 25.7 and 30.8

**Table 6. Ovarian markers in childhood cancer survivors (CCSs) (n = 167) and controls (n = 164).**

| Group | 1 | 2 | 3 | 4 | All CCSs | Controls | *p*-value All CCSs vs controls | Overall *p*-value |
|---|---|---|---|---|---|---|---|---|
| N | 54 | 13 | 48 | 50 | 167 | 164 | n.a. | n.a. |
| Age at examination (years) | 30.6 (19.6–49.9) | 39.4 (25.0–54.3) | 35.1 (19.9–49.8) | 37.2 (19.3–57.8) | 34.6 (19.3–57.8) | 35.8 (19.3–58.0) | 0.385 | 0.053 |
| Age at diagnosis (years) | 6.2 (0.1–17.9) | 12.6 (4.6–17.0) | 7.7 (0.3–17.3) | 8.5 (0.4–17.9) | 8.4 (0.1–17.9) | n.a. | n.a. | **0.011** |
| AMH (ng/ml) | 3.6 (0.0–23.5) | 0.0 (0.0–16.4) | 2.1 (0.0–11.7) | 0.4 (0.0–10.1) | 1.9 (0.0–23.5) | 2.1 (0.0–16.1) | 0.065 | **<0.001** |
| AFC | 16.5 (0–47) | n.a. | 10.0 (0–38) | 7.0 (0–27) | 12.0 (0–47) | 13.0 (0–40) | 0.096 | **<0.001** |
| OV (cm$^3$) | 8.1 (0.9–21.4) | n.a. | 6.0 (1.5–20.8) | 5.8 (0.3–21.0) | 6.8 (0.3–21.4) | 8.0 (0.9–36.7) | **0.021** | **0.004** |

CCSs were divided into infertility risk groups according to PanCareLIFE recommendations based on cancer treatment given (Table 2).

Two CCSs with bilateral oophorectomy were not included.

years, respectively (*p* = 0.048). Regarding age at diagnosis, the Kruskal-Wallis test showed a significant difference (*p* = 0.010), but pairwise comparisons did not. The levels of AMH and AFC were significantly reduced in group 4 compared with controls (*p* = 0.004 and *p*<0.001, respectively) (Fig 2d and 2e). Furthermore, OV was significantly lower both in groups 3 and 4 compared with controls (*p* = 0.043 and *p* = 0.033, respectively) (Fig 2f).

## Fertility

A trend towards reduced fertility for all CCSs was observed when compared with controls, with fewer women achieving pregnancies and giving birth. However, no statistically significant differences were seen regarding fertility data, although more CCSs underwent fertility investigation and treatment. Seven CCSs and four controls had ovarian stimulation, eight CCSs and eight controls underwent in vitro fertilisation (IVF), six CCSs had egg donation though none of the controls received this, and four CCSs and one control adopted a child. Of 18 CCSs who received fertility treatment, 11 (61%) gave birth. It should be noted that CCSs (n = 72) were somewhat younger than controls (n = 87) when giving birth to their first child: median age

**Table 7. Ovarian markers in childhood cancer survivors (CCSs) (n = 120) and controls (n = 113) <40 years.**

| Group | 1 | 2 | 3 | 4 | All CCSs <40 years | Controls <40 years | *p*-value All CCSs vs controls <40 years | Overall *p*-value |
|---|---|---|---|---|---|---|---|---|
| N | 40 | 7 | 38 | 35 | 120 | 113 | n.a. | n.a. |
| Age at examination (years) | 25.7 (19.6–38.3) | 38.3 (25.0–39.4) | 33.3 (19.9–39.4) | 30.7 (19.3–39.8) | 30.6 (19.3–39.8) | 30.8 (19.3–39.9) | 0.599 | **0.005** |
| Age at diagnosis (years) | 4.9 (0.1–15.1) | 12.5 (4.6–15.0) | 7.2 (0.3–17.2) | 7.9 (0.4–17.2) | 6.4 (0.1–17.2) | n.a. | n.a. | **0.010** |
| AMH (ng/ml) | 4.0 (0.1–23.5) | 0.1 (0.0–16.4) | 2.9 (0.0–11.7) | 0.5 (0.0–9.4) | 3.3 (0.0–23.5) | 3.2 (0.0–16.1) | 0.108 | **<0.001** |
| AFC | 20.0 (4–47) | n.a. | 11.0 (3–38) | 7.0 (0–27) | 13.0 (0–47) | 15.0 (5–40) | **0.026** | **<0.001** |
| OV (cm$^3$) | 10.1 (1.7–21.4) | n.a. | 6.1 (1.5–20.8) | 6.0 (0.5–21.0) | 7.6 (0.25–21.4) | 9.1 (1.6–36.7) | **0.009** | **0.002** |

CCSs were divided into infertility risk groups according to PanCareLIFE recommendations based on cancer treatment given (Table 2).

Data presented as count, median, and range.

AMH: anti-Müllerian hormone; AFC: antral follicle count; OV: ovarian volume; n.a.: not applicable.

*P*-value for all CCSs vs controls calculated with Mann-Whitney U test.

Overall *p*-value calculated with Kruskal-Wallis test.

**Table 8. Fertility outcome for childhood cancer survivors (CCSs) (n = 167) and controls (n = 164).**

| | No risk | Low-risk | Moderate-risk | High-risk | Very high-risk | Only surgery | Unilateral oophorectomy | All CCSs | Controls | *p*-value All CCSs vs controls | Overall *p*-value |
|---|---|---|---|---|---|---|---|---|---|---|---|
| N | 17 | 31 | 29 | 37 | 13 | 19 | 13 | 167 | 164 | n.a. | n.a. |
| Age at examination (years) | 32.9 (19.6–49.8) | 31.8 (19.8–48.0) | 28.8 (19.5–47.9) | 36.6 (19.3–57.8) | 30.8 (21.8–51.4) | 38.1 (22.2–49.9) | 39.4 (25.0–54.3) | 34.6 (19.3–57.8) | 35.8 (19.3–58.0) | 0.385 | **0.015** |
| Age at diagnosis (years) | 11.1 (1.3–17.9) | 5.0 (0.6–16.0) | 6.1 (0.1–17.2) | 7.9 (0.4–17.6) | 12.3 (1.8–17.9) | 8.8 (0.1–17.2) | 12.6 (4.6–17.0) | 8.4 (0.1–17.9) | n.a. | n.a. | **0.003** |
| POI, n | 0 | 1 (3%) | 0 | 5 (14%) | 9 (69%) | 0 | 4 (31%) | 22 (13%) | 0 | **<0.001** | n.a. |
| Females pregnant, n | 6 (35%) | 13 (42%) | 11 (38%) | 22 (59%) | 3 (23%) | 14 (74%) | 9 (69%) | 84 (50%) | 97 (59%) | 0.122 | **0.010** |
| Pregnancies, total n | 10 | 37 | 34 | 49 | 5 | 43 | 32 | 224 | 263 | n.a. | n.a. |
| Pregnancies/ female, n | 0 (0–3) | 0 (0–6) | 0 (0–6) | 1 (0–5) | 0 (0–3) | 2 (0–11) | 2 (0–7) | 1 (0–11) | 1 (0–7) | 0.066 | **0.005** |
| Females given birth, n | 5 (29%) | 11 (35%) | 9 (31%) | 21 (57%) | 2 (15%) | 13 (68%) | 7 (54%) | 74 (44%) | 88 (54%) | 0.100 | **0.006** |
| Children born, total n | 8 | 23 | 18 | 40 | 3 | 29 | 19 | 152 | 180 | n.a. | n.a. |
| Children born/ female, n | 0 (0–3) | 0 (0–3) | 0 (0–4) | 1 (0–4) | 0 (0–2) | 1 (0–5) | 2 (0–4) | 0 (0–5) | 1 (0–5) | 0.117 | **0.006** |
| Children born/ pregnancies | 80% | 62% | 53% | 82% | 60% | 67% | 59% | 68% | 68% | n.a. | n.a. |
| Fertility investigation, n | 3 (18%) | 0 | 2 (7%) | 5 (14%) | 4 (31%) | 4 (21%) | 3 (23%) | 25 (15%) | 15 (9%) | 0.129 | n.a. |
| Fertility treatment, n | 1 (6%) | 0 | 2 (7%) | 3 (8%) | 2 (15%) | 3 (16%) | 3 (23%) | 18 (11%) | 12 (7%) | 0.339 | n.a. |
| Adoption | 0 | 0 | 0 | 0 | 2 (15%) | 0 | 1 (8%) | 4 (2%) | 1 (1%) | 0.372 | n.a. |

CCSs were divided into Swedish infertility risk groups according to cancer treatment given (Table 1).

For six CCSs, risk assessment was not available; two with bilateral oophorectomy were not included.

Data presented as count, median, range, and percentage.

POI: premature ovarian insufficiency; n.a.: not applicable.

*P*-value for all CCSs vs controls calculated with Mann-Whitney U test and Fisher's exact test.

Overall *p*-value calculated with Kruskal-Wallis test and chi-squared test.

26.5 and 29.0 years, respectively (*p* = 0.013). Furthermore, there was no significant association between future desire for children and the groups at all ages (*p* = 0.883).

For the Swedish infertility risk groups there was a difference regarding number of pregnancies (*p* = 0.005) and children born (*p* = 0.006), but pairwise comparisons showed no differences. Fewer women were able to get pregnant in the very high-risk group compared with controls, 23% vs. 59%, respectively. The same trend was observed for childbirth, with two (15%) women only giving birth to a total of three children in the very high-risk group. In addition, more women in the very high-risk group as well as in the unilateral oophorectomy group underwent fertility investigation and treatment (Table 8).

There were no significant differences based on the PanCareLIFE recommendations regarding number of pregnancies and children born across the groups at all ages (Table 9). However, there was a trend towards impaired fertility for women of all ages, with fewer being able to get pregnant with a median of 0.0 for number of pregnancies per female in both groups 3 and 4. This trend was also apparent for giving birth, with a median of 0.0 children born per female in

**Table 9. Fertility outcome for childhood cancer survivors (CCSs) (n = 167) and controls (n = 164).**

| Group | 1 | 2 | 3 | 4 | All CCSs | Controls | *p*-value All CCSs vs controls | Overall *p*-value |
|---|---|---|---|---|---|---|---|---|
| N | 54 | 13 | 48 | 50 | 167 | 164 | n.a. | n.a. |
| Age at examination (years) | 30.6 (19.6–49.9) | 39.4 (25.0–54.3) | 35.1 (19.9–49.8) | 37.2 (19.3–57.8) | 34.6 (19.3–57.8) | 35.8 (19.3–58.0) | 0.385 | 0.053 |
| Age at diagnosis (years) | 6.2 (0.1–17.9) | 12.6 (4.6–17.0) | 7.7 (0.3–17.3) | 8.5 (0.4–17.9) | 8.4 (0.1–17.9) | n.a. | n.a. | **0.011** |
| POI, n | 1 (2%) | 4 (31%) | 0 | 15 (30%) | 22 (13%) | 0 | **<0.001** | **<0.001** |
| Females pregnant, n | 28 (52%) | 9 (69%) | 22 (46%) | 24 (48%) | 84 (50%) | 97 (59%) | 0.122 | 0.285 |
| Pregnancies, total n | 81 | 32 | 54 | 56 | 224 | 263 | n.a. | n.a. |
| Pregnancies/female, n | 1 (0–11) | 2 (0–7) | 0 (0–6) | 0 (0–5) | 1 (0–11) | 1 (0–7) | 0.066 | 0.071 |
| Females given birth, n | 25 (46%) | 7 (54%) | 19 (40%) | 22 (44%) | 74 (44%) | 88 (54%) | 0.100 | 0.419 |
| Children born, total n | 53 | 19 | 35 | 44 | 152 | 180 | n.a. | n.a. |
| Children born/female, n | 0 (0–5) | 2 (0–4) | 0 (0–4) | 0 (0–4) | 0 (0–5) | 1 (0–5) | 0.117 | 0.279 |
| Children born/ pregnancies | 65% | 59% | 65% | 79% | 68% | 68% | n.a. | n.a. |
| Fertility investigation, n | 4 (7%) | 3 (23%) | 6 (13%) | 10 (20%) | 25 (15%) | 15 (9%) | 0.129 | n.a. |
| Fertility treatment, n | 3 (6%) | 3 (23%) | 3 (6%) | 7 (14%) | 18 (11%) | 12 (7%) | 0.339 | n.a. |
| Adoption | 0 | 1 (8%) | 0 | 2 (4%) | 4 (2%) | 1 (1%) | 0.372 | n.a. |

CCSs were divided into infertility risk groups according to PanCareLIFE recommendations based on cancer treatment given (Table 2).

Two CCSs with bilateral oophorectomy were not included.

Data presented as count, median, range, and percentage.

POI: premature ovarian insufficiency; n.a.: not applicable.

*P*-value for all CCSs vs controls calculated with Mann-Whitney U test and Fisher's exact test.

Overall *p*-value calculated with Kruskal-Wallis test and chi-squared test.

groups 1, 3, and 4. In addition, fertility investigation and treatment, as well as adoption, were more frequent among women in groups 2 and 4 compared with controls.

## POI

A total of 22 (13%) CCSs met the criteria of POI. Of those, six had primary amenorrhea and 15 secondary amenorrhea with either undetectable (<0.023 ng/ml) or very low AMH (<0.1 ng/ml). One CCS with primary amenorrhoea and ongoing HRT was also included in the POI group despite AMH of 0.403 ng/ml.

For the Swedish infertility risk groups, CCSs with POI were present in the very high-risk group (9/13), the high-risk group (5/37), the unilateral oophorectomy group (4/13), and one in the low-risk group (1/31) compared with none among controls (Table 8). Furthermore, POI was observed in 30%, 31%, and 2% among CCSs in the PanCareLIFE groups 4, 2, and 1, respectively (Table 9).

## Discussion

We found that females diagnosed and treated for cancer in childhood have somewhat reduced AMH levels in adulthood compared with healthy controls, although this was not significant. Additionally, no difference was observed in AMH levels for females below 40 years of age. However, subgroup analyses of females <40 years based on the Swedish recommendations, showed that both the very high-risk group and the high-risk group had lower AMH and AFC, but no significant difference was noted concerning OV compared with controls. AMH and

AFC were likewise significantly reduced in the PanCareLIFE group 4 in comparison with controls. Moreover, POI was primarily observed in the very high-risk group (69%), the high-risk group (14%), and in the unilateral oophorectomy group (31%). We also observed POI in 30% and 31% of CCSs in groups 4 and 2, respectively. A Swedish study with younger CCSs (i.e., 19–40 years of age at study) reported POI in 9% of survivors, with a prevalence of 35% for those receiving highly gonadotoxic treatments [26]. In comparison, the total prevalence of POI is 1.9% in Sweden [27]. Our findings are concordant with previous studies [28–30] confirming that high-doses of alkylating agents, radiotherapy to the ovaries and HSCT are the most gonadotoxic treatments. There has been a discussion regarding CED cut-off, 6 or 8 $g/m^2$, that should be recommended for fertility preservation [17]. We believe our study supports the limit of 6 $g/m^2$.

For CCSs of all ages classified as very high-risk for infertility we noticed a trend towards impaired fertility, with fewer women being able to get pregnant and to give birth. This tendency was also observable for those in groups 3 and 4, as well as for all CCSs compared with controls. Moreover, fertility investigation and treatment were more common among CCSs, especially for those who received highly gonadotoxic treatment (i.e., the very high-risk group and group 4) or underwent unilateral oophorectomy. Previous studies similarly reported reduced fertility in CCSs [31, 32]. A study by van den Berg et al., identified that CCSs have a significantly increased risk of impaired fertility with a CED >7121 $mg/m^2$ and at any radiotherapy dose to the lower abdomen, which is consistent with our findings in group 4 [33]. It should be mentioned that the cut-off for low radiotherapy dose to lower abdomen was set rather high (>0–≤24 Gy) in this study by van den Berg et al. Moreover, Dillon et al. reported that pregnancies were achieved at a similar rate in young cancer survivors with reduced AMH and AFC [34]. We also found significantly lower AMH levels and AFC in the high-risk group for those below 40 years with no sign of affected fertility in terms of number of pregnancies and children born. One theory for these observations could be that ovarian markers reflect the quantity rather than the quality of the remaining follicles. In addition, a recent study by Harris et al. stated that AMH is a poor predictor of future fertility [35]. Another explanation could be earlier childbearing for CCSs after receiving information and recommendations from health care providers during follow-up. Indeed, the CCSs in the current study were somewhat younger than controls when giving birth to their first child, with median age 26.5 and 29.0 years, respectively. Furthermore, no difference was observed between CCSs and controls regarding desire for future children, which could be interpreted as already having had the desired number of children or reaching an acceptance of not being able to have biological children. Considering that women in Europe have their first child at an older age, it is of importance to follow AMH in CCSs at risk after treatment, as they might have a shortened reproductive lifespan.

The unilateral oophorectomy group is heterogeneous regarding cancer treatments given, which raises complexity when interpreting the results. In addition, three CCSs in this group had one ovary removed independent of their cancer diagnosis. Median AMH levels in this group for those below 40 years of age were low (0.1 ng/ml) when compared with controls (3.2 ng/ml), although not significant, and the prevalence of POI was 31%. However, there was no trend noted towards impaired fertility (i.e., achieved number of pregnancies and children born), but this group had the highest use of fertility treatment (23%). Keeping in mind the limited number of females included in the unilateral oophorectomy group, future studies need to evaluate whether this group is also at risk of affected fertility.

To our knowledge, this is the first study evaluating the Swedish infertility risk classification and the PanCareLIFE recommendations in a patient cohort treated before fertility preservation was established among childhood cancer patients. The Swedish guidelines are more comprehensive compared with the PanCareLIFE recommendations, dividing the PanCareLIFE

group 4 into the very high-risk and the high-risk groups (Tables 1 and 2) [17, 18]. However, both guidelines strongly recommend oocyte cryopreservation for post-pubescent girls in these groups before start of cancer treatment. The Swedish recommendations also propose ovarian tissue cryopreservation for pre-pubescent girls with very high-risk, while the PanCareLIFE only moderately recommends this measure for pre-pubescent girls in group 4. It should be mentioned that these recommendations are considered only when not delaying treatment and compromising prognosis. Furthermore, the Swedish guidelines do not consider cranial radiotherapy for infertility risk assessment. Based on the PanCareLIFE guidelines, all women treated with cranial radiotherapy, as well as CED $<6$ g/m$^2$, were assigned to group 3. As previously mentioned, this group presented a trend towards reduced fertility without significantly lower AMH and AFC. In the Swedish guidelines the moderate-risk group could be correlated to group 3 apart from cranial radiotherapy. However, the moderate-risk group showed neither reduced ovarian markers nor impaired fertility. It could therefore be argued that the observed trend towards fertility impairment in group 3 is due to hypogonadotropic hypogonadism rather than an ovarian insult. In case of impact to the hypothalamic–pituitary axis, hormonal treatment for ovarian stimulation can be offered when pregnancy is desired.

Previous studies report that the insult to ovarian function seems to occur once regardless of treatment modality and does not induce accelerated ovarian loss. This assertion is limited by a short follow-up of three years [36, 37]. In our study, a small difference in AMH levels was the same at all ages between CCSs and controls, supporting the theory of one hit at cancer treatment with no further follicle loss. Future prospective longitudinal studies are needed to confirm these results.

Our study has several limitations. We could not predict fertility outcomes (i.e., number of pregnancies and children born) and POI with ovarian reserve markers because of the cross-sectional design. The number of study participants was limited in the infertility risk groups. Some women were young at study enrolment and had not reached mean age for a first born child in Sweden, which was 29.1 years in 2015 [38]. As data were collected between 2010 and 2015, it caused a difference of zero to three years for gathering of blood samples between CCSs and controls. Furthermore, studies have reported a reduction of AMH levels by use of oral hormonal therapy, while others have not, posing a possible limitation to our study, although the number of CCSs and controls on OC or HRT was comparable [39–41]. The transvaginal ultrasound examination was conducted by six doctors, which might contribute to interindividual differences. There were also difficulties assessing AFC and OV for CCSs, as 19% and 23% respectively, could not be estimated. The radiotherapy dose was in most cases reported as an interval rather than the exact dose. At last, we were unable to identify the gonadotoxic doses of radiotherapy and each chemotherapeutic agent due to heterogeneous treatments, multiple diagnoses, and limited sample size.

This study's strengths were careful examination of the women for evaluation of ovarian function with ultrasound and serum markers and detailed treatment data for chemotherapeutic agents and radiotherapy doses for CCSs. Data was compared with a matched control group. We also had information about fertility for those who declined participation and could verify that the final study group of CCSs was not different regarding offspring.

Future prospective studies are required to reveal the predictive value of ovarian markers for fertility outcomes in CCSs. Studies with longer follow-up are needed to assess the reduction rate of ovarian markers over time and to prognosticate time of POI occurrence. New therapeutic options, like use of Inositols (i.e., insulin-sensitizing compounds) for assisted reproduction, might improve the chance of obtaining qualitatively good oocytes from CCSs at risk of ovarian dysfunction [42]. There is also a need for studies to identify genetic markers indicative of

susceptibility to chemotherapy induced gonadal toxicity in order to give individualised fertility preservation recommendations.

In conclusion, POI and reduced AMH levels, as well as a trend towards fertility impairment, were associated with treatment burden according to the Swedish and the PanCare-eLIFE infertility risk classifications. This supports treatments including CED $\geq$6 g/m$^2$, radiotherapy exposure to ovaries or HSCT as being the most gonadotoxic. Patients at risk and their families should be provided with fertility preservation counselling before initiation of cancer treatment and follow-up at a young reproductive age.

## Supporting information

**S1 Table. Cancer diagnoses and treatments.**
(DOCX)

**S1 File. Linear regression analyses.**
(DOCX)

## Acknowledgments

The authors thank the research nurses Irene Leijonhufvud and Lena Rollof for their expert data collection and supportive services during this study. We also thank Thomas Wiebe for establishing BORISS and providing us with valuable data. Lastly, we would like to express our gratitude to Professor Aleksander Giwercman, the previous principal investigator who initiated the study and was responsible for organizing inclusion and investigation of study subjects.

## Author Contributions

**Conceptualization:** Anna Nyström, Helena Mörse, Maria Elfving.

**Data curation:** Anna Nyström, Maria Elfving.

**Formal analysis:** Anna Nyström, Maria Elfving.

**Funding acquisition:** Maria Elfving.

**Investigation:** Anna Nyström, Helena Mörse, Emir Henic, Maria Elfving.

**Methodology:** Anna Nyström, Maria Elfving.

**Project administration:** Anna Nyström, Maria Elfving.

**Resources:** Jacob Engellau, Elinore Wieslander, Andrzej Tomaszewicz.

**Supervision:** Helena Mörse, Ingrid Øra, Emir Henic, Maria Elfving.

**Validation:** Anna Nyström, Helena Mörse, Ingrid Øra, Emir Henic, Maria Elfving.

**Visualization:** Anna Nyström, Maria Elfving.

**Writing – original draft:** Anna Nyström.

**Writing – review & editing:** Anna Nyström, Helena Mörse, Ingrid Øra, Emir Henic, Jacob Engellau, Elinore Wieslander, Andrzej Tomaszewicz, Maria Elfving.

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
