## [Decision Letter · Decision Letter 0]

2 Jun 2024

PONE-D-24-15437Anti-Müllerian hormone and fertility in women after childhood cancer treatment: association with current infertility risk classificationsPLOS ONE

Dear Dr. Nyström,

Thank you for submitting your manuscript to PLOS ONE. After careful consideration, we feel that it has merit but does not fully meet PLOS ONE’s publication criteria as it currently stands. Therefore, we invite you to submit a revised version of the manuscript that addresses the points raised during the review process.

We look forward to receiving your revised manuscript.

Kind regards,

Antoine Naem, M.D.

Academic Editor

PLOS ONE

Journal Requirements:

2. In the online submission form, you indicated that [The data underlying this article is not publicly available due to potentially identifying and sensitive information (e.g., specific dates, cancer diagnoses and treatments, etc.). It will be shared on reasonable request to Lund University (contact via ikvl-prefektadmin@med.lu.se).]. 

4. We note that you have referenced (unpublished) on page 8, which has currently not yet been accepted for publication. Please remove this from your References and amend this to state in the body of your manuscript: (ie “Bewick et al. [Unpublished]”) as detailed online in our guide for authors

Reviewers' comments:

Reviewer's Responses to Questions

**Comments to the Author**

1. Is the manuscript technically sound, and do the data support the conclusions?

Reviewer #1: Yes

Reviewer #2: Partly

Reviewer #3: Yes

2. Has the statistical analysis been performed appropriately and rigorously? 

Reviewer #1: Yes

Reviewer #2: I Don't Know

Reviewer #3: Yes

3. Have the authors made all data underlying the findings in their manuscript fully available?

Reviewer #1: Yes

Reviewer #2: Yes

Reviewer #3: No

4. Is the manuscript presented in an intelligible fashion and written in standard English?

Reviewer #1: No

Reviewer #2: Yes

Reviewer #3: Yes

5. Review Comments to the Author

Reviewer #1: I read with great interest the Manuscript, which falls within the aim of this Journal.

In my honest opinion, the topic is interesting enough to attract the readers’ attention. Methodology is accurate and conclusions are supported by the data analysis. Nevertheless, authors should clarify some point and improve the discussion citing relevant and novel key articles about the topic.

Authors should consider the following recommendations:

- Manuscript should be further revised by a native English speaker.

- Inclusion/exclusion criteria should be better clarified.

- The authors have not adequately highlighted the strengths and limitations of their study. I suggest clarifying these points.

- What are the actual clinical implications of this study? it is important to report the results obtained by the authors in the context of clinical practice and to adequately highlight what contribution this study adds to the literature already existing on the topic and to future study perspectives.

- The discussion should be further expanded, referring to the following PMID: 37798243, briefly discussing how the use of inositols can increase the chances of obtaining quantitatively and qualitatively good oocytes.

Reviewer #2: I have read with great interest the proposed manuscript. It is well written and it addresses a very important topic. With this being said I have failed to see what is new and what this brings to the field in 2024. I think everyone agrees with the conclusions.

Reviewer #3: This study encompasses an interesting study cohort with sound data, especially considering the usual caveats for these long term follow-up cohorts. Its findings strengthen the current knowledge regarding different risk groups, and shows that the ovarian function/reserve is lower in the groups with greater risk. It is an interesting descriptions of their cohort. I’d like to challenge the group to also investigate what the accuracy is of the evaluated risk groups: perhaps using an ROC or sensitivity/specificity using one or more endpoints. Perhaps they can identify a cluster of women who showed a different outcome than expected based on their treatment risk group?

Minor details:

Methods: were there no women with a natural but irregular cycle? Were they excluded? Were there more outliers perhaps in any of the groups regarding AMH and/of AFC?, possibly indicating PCOS?

Results page 11 line 194 : how do you know the number of offspring of the women who did not respond?

Page 12 line 219: what is the relevance of reporting that AMH levels did not differ in the groups with OC or HRT? The 20 patients on HRT must have relatively low AMH levels, if this does not lead to a difference in AMH levels between te groups one wonders a) why combine it with a completely different group on OC and b) if the power is anywhere near giving it an interesting significance.

The results stated in this paragraph are not depicted in a table? That can be a choice, perhaps mentioning this.

6. PLOS authors have the option to publish the peer review history of their article (what does this mean?). If published, this will include your full peer review and any attached files.

Reviewer #1: No

Reviewer #2: No

Reviewer #3: No

---

## [Author Response · Author response to Decision Letter 0]

24 Jun 2024

Please see the uploaded file “Response to Reviewers’”, where the journal requirements as well as reviewers' comments are addressed.

---

## [Decision Letter · Decision Letter 1]

30 Jul 2024

Anti-Müllerian hormone and fertility in women after childhood cancer treatment: association with current infertility risk classifications

PONE-D-24-15437R1

Dear Dr. Nyström,

We’re pleased to inform you that your manuscript has been judged scientifically suitable for publication and will be formally accepted for publication once it meets all outstanding technical requirements.

Kind regards,

Antoine Naem, M.D.

Academic Editor

PLOS ONE

Reviewers' comments:

Reviewer's Responses to Questions

**Comments to the Author**

1. If the authors have adequately addressed your comments raised in a previous round of review and you feel that this manuscript is now acceptable for publication, you may indicate that here to bypass the “Comments to the Author” section, enter your conflict of interest statement in the “Confidential to Editor” section, and submit your "Accept" recommendation.

Reviewer #1: (No Response)

Reviewer #2: All comments have been addressed

Reviewer #3: All comments have been addressed

2. Is the manuscript technically sound, and do the data support the conclusions?

Reviewer #1: (No Response)

Reviewer #2: Partly

Reviewer #3: Yes

3. Has the statistical analysis been performed appropriately and rigorously? 

Reviewer #1: (No Response)

Reviewer #2: I Don't Know

Reviewer #3: Yes

4. Have the authors made all data underlying the findings in their manuscript fully available?

Reviewer #1: (No Response)

Reviewer #2: Yes

Reviewer #3: Yes

5. Is the manuscript presented in an intelligible fashion and written in standard English?

Reviewer #1: (No Response)

Reviewer #2: Yes

Reviewer #3: Yes

6. Review Comments to the Author

Reviewer #1: (No Response)

Reviewer #2: (No Response)

Reviewer #3: (No Response)

7. PLOS authors have the option to publish the peer review history of their article (what does this mean?). If published, this will include your full peer review and any attached files.

Reviewer #1: No

Reviewer #2: No

Reviewer #3: No

---

## [Editor Report · Acceptance letter]

1 Aug 2024

PONE-D-24-15437R1 

PLOS ONE

Dear Dr. Nyström, 

I'm pleased to inform you that your manuscript has been deemed suitable for publication in PLOS ONE. Congratulations! Your manuscript is now being handed over to our production team.

Kind regards, 

on behalf of

Dr. Antoine Naem 

Academic Editor

PLOS ONE